# An Investigation of the Thermal Transitions and Physical Properties of Semiconducting PDPP4T:PDBPyBT Blend Films

**DOI:** 10.3390/ma15238392

**Published:** 2022-11-25

**Authors:** Barbara Hajduk, Paweł Jarka, Tomasz Tański, Henryk Bednarski, Henryk Janeczek, Paweł Gnida, Mateusz Fijalkowski

**Affiliations:** 1Centre of Polymer and Carbon Materials, Polish Academy of Sciences, 34 Marie Curie-Skłodowska Str., 41-819 Zabrze, Poland; 2Department of Engineering Materials and Biomaterials, Silesian University of Technology, 18a Konarskiego Str., 41-100 Gliwice, Poland; 3Institute for Nanomaterials, Advanced Technologies and Innovation, Technical University of Liberec, Studentská 1402/2, 461 17 Liberec, Czech Republic

**Keywords:** variable-temperature spectroscopic ellipsometry, polymer blend films, organic semiconductors

## Abstract

This work focuses on the study of thermal and physical properties of thin polymer films based on mixtures of semiconductor polymers. The materials selected for research were poly [2,5-bis(2-octyldodecyl)-pyrrolo [3,4-c]pyrrole-1,4(2H,5H)-dione-3,6-diyl)-alt-(2,2′;5′,2″;5″,2′′′-quater-thiophen-5,5′′′-diyl)]—PDPP4T, a p-type semiconducting polymer, and poly(2,5-bis(2-octyldodecyl)-3,6-di(pyridin-2-yl)-pyrrolo [3,4-c]pyrrole-1,4(2H,5H)-dione-alt-2,2′-bithiophene)—PDBPyBT, a high-mobility n-type polymer. The article describes the influence of the mutual participation of materials on the structure, physical properties and thermal transitions of PDPP4T:PDBPyBT blends. Here, for the first time, we demonstrate the phase diagram for PDPP4T:PDBPyBT blend films, constructed on the basis of variable-temperature spectroscopic ellipsometry and differential scanning calorimetry. Both techniques are complementary to each other, and the obtained results overlap to a large extent. Our research shows that these polymers can be mixed in various proportions to form single-phase mixtures with several thermal transitions, three of which with the lowest characteristic temperatures can be identified as glass transitions. In addition, the RMS roughness value of the PDPP4T:PDBPyBT blended films was lower than that of the pure materials.

## 1. Introduction

Organic electronics (with particular emphasis on optoelectronics and photovoltaics) is currently the target of extensive research carried out by research units around the world. The specific interest is the use of semiconductor polymer thin films in the production of organic electronic devices (OEDs), such as electroluminescent diodes OLEDs, solar cells (OPVs) or field effect transistors (OFETs) [1,2,3,4].

When used in electronic systems, organic semiconductors, such as oligomers, conjugated polymers or molecular materials in the form of active layers, offer an advantage over the use of inorganic materials, such as the possibility of obtaining multifunctional large-surface flexible lightweight devices using fast and economical printing techniques e.g., screen printing, doctor blade, spin coating or roll-to-roll processing [5,6].

Ambipolar polymers containing an alternating donor (D) and acceptor (A) in their repeating unit have been successfully used as a single-component semiconductor OFET manufacturing technology [7]. These polymers are used in the CMOS (complementary metal oxide) technology presenting a low power demand, high speed of switching, very good signal robustness and high immunity of noise [8,9]. At the same time, conjugate polymers are widely used in high efficiency organic photovoltaic cells (OPV). The use of this type of material is associated with a wide range of solar radiation absorption, high mobility of the carrier and improved “face-on” orientation of polymers coupled in thin layers [10,11,12,13]. On the other hand, good OFET performance requires “edge on” orientation of the conjugated polymers in the thin films.

An ambipolar polymer based on a diketopyrrolopyrrole DPP (DBPy) unit as an acceptor has attracted particular interest due to the location of the LUMO level below –4 eV and the efficient electron injection and charge transport (reported electron and hole mobility up to 6.3 cm^2^·V^−1^·s^−1^ and 2.78 cm^2^·V^−1^·s^−1^, respectively) [14,15]. From reports published by Zhao et al. and Gao et al., the application of the DPP-based donor–acceptor heterojunction may allow the achievement of solar cell performance (PCE) up to 12% [16,17]. Liu et al. [18] obtained an efficiency of 7.59% (V_OC_ of 0.61 V, JSC of 17.95 mA/cm^2^, and FF of 69.6%) for a device based on a heterojunction containing crystalline diketopyrrolopyrrole (DPP) polymer and PC71BM [18].

The performance of OEDs can be optimized in many ways, for example, by changing the volume composition of the active layer [19,20]. Different methods include the annealing process [21,22,23] or the use of different solvents [24]. Another method of improving the efficiency is the extraction/injection of electron layers and/or holes in the area of their contact with the organic semiconductor [25,26,27]. A good example of work that uses the annealing process to optimize the performance of OPV devices based on the layer of the P3HT:PCBM blend is the work of Pearson et al. [20]. This work shows that the efficiency of the OPV increases when the prepared devices were annealed at a temperature higher than the upper glass transition temperature Tg. This example clearly shows that the upper Tg is a very important technological parameter of the OPV preparation process. Nevertheless, understanding the origin of all glass transition temperatures, including the lower ones, can be very important in the further development of organic optoelectronics [28].

It turns out, as in our earlier works [29,30,31,32], that the variable-temperature spectroscopic ellipsometry (VTSE) non-destructive optical technique is very sensitive to thermal transitions and is very complementary to differential scanning calorimetry (DSC) with regard to the examination of organic layers. Literature data show that the identification of the characteristic temperatures of thermal transitions can be made on the basis of raw ellipsometric data [33,34,35,36,37,38,39,40,41] and additionally confirmed by an appropriate analysis of layer thickness changes caused by temperature change or various physical parameters [42,43,44,45,46,47,48,49,50].

Here we present an in-depth study of the influence of the composition of PDPP4T:PDBPyBT blends films on their thermal transitions using VTSE and DSC measurement techniques. For this research, we chose PDPP4T as the organic p-type semiconductor (polymer donor) and PDBPyBT as the n-type semiconductor (polymer acceptor). The electronic structure of these polymers is described in refs. [3,51].

An important reason for selecting these particular polymers is that they share the same aliphatic side chains and aromatic rings in the structural unit of the polymer backbone. It can, therefore, be expected that these polymers will be compatible with each other and mix well. To the best of our knowledge, we present, for the first time, the PDPP4T:PDBPyBT blended film phase diagram based on both of these methods. Our research clearly shows that these polymers can be mixed in various proportions to form single-phase mixtures with several thermal transitions, three of which with the lowest characteristic temperatures can be identified as glass transitions [28]. Moreover, the RMS roughness of the films of the PDPP4T:PDBPyBT blends turned out to be lower than that of the polymer blends. 

## 2. Experimental

The materials used were poly [2,5-bis(2-octyldodecyl)pyrrolo [3,4-c]pyrrole-1,4(2H,5H)-dione-3,6-diyl)-alt-(2,2′;5′,2″;5″,2′′′-quaterthiophen-5,5′′′-diyl)]—M333-PDPP4T [16,18,52,53], with a molar mass Mw = 84 kDa, and poly(2,5-bis(2-octyldodecyl)-3,6-di(pyridin-2-yl)-pyrrolo[3,4-c]pyrrole-1,4(2H,5H)-dione-alt-2,2′-bithiophene)—M323-PDBPyBT [7,14], with a molar mass Mw = 114 kDa. The materials were supplied by Ossila. Their chemical structures are shown in Figure 1.

To prepare the samples, the polymeric material was dissolved in chloroform (Aldrich, 98.5% pure, St. Louis, MO, USA). The weight concentration of all prepared solutions was 10 mg/mL (the percentage composition of the solutions is shown in Table 1). Additionally, all solutions were homogenized using a Bandelin Sonoplus homogenizer at 16 kJ for 10 min. Using the polymer solutions prepared in this way, the layers were deposited by spin coating and drop casting on microscope slides and silicone (with a 400 nm SiO_2_ layer on top) substrates; the technical parameters are presented in Table 1. In the next stage, the prepared films were annealed at 120 °C for 10 min in order to remove solvent residues. All samples were stored in a dry laboratory box at room temperature. A dry box with a rubber seal was half-filled with a hygroscopic gel and kept in a glovebox under a nitrogen atmosphere.

All ellipsometric measurements were made using a SENTECH SE 850E ellipsometer, working under SpectraRay/3 software, which operates in a 240–2500 nm spectral range. Variable-temperature measurements were performed using a temperature chamber operating at low vacuum (10^−1^ Tr). The temperature was controlled by an INSTEC mK 1000 controller, using an electrical heater and liquid nitrogen circuit. We used a standard quenching protocol, in which the value of the maximum heating temperature depends on the type of material tested. Accordingly, each sample was heated around its melting point for 2 min and then cooled to −50 °C at an average cooling rate of 100 °C/min. The VTSE study of thermal transitions was carried out in the wavelength range of 240–930 nm. The time interval between single measurements was 10 s. Measurements were made in heating mode at a rate of 2.0 °C/min. The thickness of the obtained layers was determined in the variable-angle ellipsometry mode. Ellipsometric angles Ψ and Δ were measured in the UV/Vis range of 240–930 nm. The angle of incidence of the light beam was in the range of 40–70°, and single measurements were made in steps of 2.5°. Optical absorption spectra were also recorded using the same ellipsometer operating in transmission mode with the goniometer set at 90° and equipped with a special sample holder.

Differential scanning calorimetry measurements were performed using the Q2000 apparatus (TA Instruments, Newcastle, DE, USA), with aluminum sample pans. Thermal characteristics of the samples were obtained under nitrogen atmosphere (gas flow 50 mL/min). The instrument was calibrated with high-purity indium standards. DSC measurements were performed on original powder materials from the supplier and obtained from very thick films (about 2 μm), which were removed from glass substrates. The cooling and heating rate was 20 °C/min.

X-ray diffraction (XRD) scans were performed using a D8 Advance diffractometer (Bruker, Karlsruhe, Germany) with a Cu-Kα cathode (λ = 1.54 Å) in a coupled Two-Theta/Theta mode on polymer films deposited on Si/SiO_2_ substrates. The scan rate was 1.2°/min with a scanning step of 0.02° in the range of 2° to 60° for 2theta (dwell time 1 s). Background subtraction, occurring from air scattering, was performed using the DIFFRAC.EVA program.

The surface morphology was studied using the Park Systems XE 100 atomic force microscope, operated with XEI Software (Suwon, Republic of Korea). XEI enables image processing and analysis of surface roughness parameters. The microscopic measurements were taken in non-contact mode.

Films for VTSE, XRD and atomic force microscopy (AFM) studies were deposited on silicon substrates. Films for optical absorption spectra were prepared on transparent glass substrates, while films for DSC analysis were prepared on microscope slides by drop-casting and scraped off the substrate after annealing.

## 3. Results and Discussion

### 3.1. X-ray Diffraction Results

Figure 2 shows XRD scans in Bragg–Brentano geometry in the coupled two-theta/theta mode, of the tested layers in the 2theta range from 3.5 to 30 degrees, corrected for the position of the sample in relation to the X-ray beam. The clearly visible peaks in these scans for samples containing PDPP4T can be identified as coming from X-ray reflections on the crystallographic planes (100), (200), (300) and (400).

Such a conclusion is based on the observation that the respective interplanar spacing d(*hkl*), where *h*, *k* and *l* represent the Miller indices of these planes, satisfies the relationship for an orthorhombic crystal structure with axial translations, expressed by the following formula [54]:(1)$$\left(hkl\right)={\left[{\left(\frac{h}{a}\right)}^{2}+{\left(\frac{k}{b}\right)}^{2}+{\left(\frac{l}{c}\right)}^{2}\right]}^{-\frac{1}{2}},$$
where *a*, *b* and *c* are the unit cell edge lengths corresponding to the reflections from planes (100), (010) and (001), respectively. In addition, a more detailed examination of the position of the highest intensity peaks indicates a monotonic increase in their 2theta value with an increase in PDBPyBT content in the examined layers. These values, according to Bragg’s law for d(100), correspond to the following interplanar distances: 19.6, 19.8, 20.0, 20.2 and 21.0 Å, respectively, for the film with 0, 25, 50, 75 and 100% of PDBPyBT. Considering that the packing distances of polymer are revealed in thin-film diffraction scans through the presence of peaks corresponding to either co-facial π stacking (π–π spacing around 3.7 Å) or side-chain packing (lamellar spacing around 21 Å) [55,56], and the fact that in the Bragg–Brentano geometry there are visible reflections from the crystallographic planes parallel to the sample surface, from our XRD scans we can conclude that in the studied films, crystalized polymers have edge-on structures with the lamellar and the π–π stacking direction preferentially oriented parallel to the substrate. Thus, d(100) corresponds to the side-chain length. Moreover, the Scherrer equation can be used to estimate, at the simplest level, the grain size from the peak width (100) [57]. The crystallite size in the tested layers determined in this way was 243.9, 239.7, 258.2, 269.4 and 230.0 Å, respectively, for films with 0, 25, 50, 75 and 100% of PDBPyBT.

### 3.2. Transmission Measurement Results

The spectral dispersion of the absorption coefficient α of PDPP4T, PDBPyBT and their blend films is shown in Figure 3.

The optical absorption spectra were determined using ellipsometry transmission mode. A distinct absorption band extending from 900 to 550 nm corresponds to the π-π* electronic transitions and has well-distinguishable vibronic structures. The absorption bands with maxima at 450 nm can be ascribed as to coming from n-π* electronic transitions. Basing on the obtained results, we can determine the energy band gap. The values of the energy band gap Eg of pure materials and their blends were obtained using linear extrapolation to the plot (αE)^1/2^ as a function of energy E (for E > E_g_), due to the Tauc relation [58]. The Tauc relation can be used for the determination of bandgap for organic polymer semiconductors [59,60]. The obtained results show that the Eg value of PDPP4T compared to its mixture with PDBPyBT does not change significantly (Figure 4). Its value remains almost constant, even for the increasing content of PDBPyBT.

The energy bandgap of pure materials PDPP4T and PDBPyBT was 1.42 and 1.68 eV, respectively (Figure 4). In the case of the prepared blends, the value of E_g_ was 1.40 eV for the blend containing 75% of PDPP4T, 1.43 eV for the blend with a 50% content of both ingredients and 1.43 eV for the blend containing 25% of PDPP4T.

The thicknesses of layers deposited on quartz substrates were determined using the Cauchy model [29,30]:(2)$$n\left(\lambda \right)=n_{0}+C_{0}\frac{n_{1}}{\lambda^{2}}+C_{1}\frac{n_{2}}{\lambda^{4}}$$
(3)$$k\left(\lambda \right)=k_{0}+C_{0}\frac{k_{1}}{\lambda^{2}}+C_{1}\frac{k_{2}}{\lambda^{4}}$$
where *n_i_* and *k_i_*, with *i* = 0, 1 and 2, are the model (fitting) parameters and the coefficients *C*_0_ and *C*_1_ are the numerical constants. The determined thickness values are presented in Table 2. 

### 3.3. Thermal Analysis Results

Thermal analysis was performed in two ways: using variable-temperature spectroscopic ellipsometry and differential scanning calorimetry. As in our previous works [29,30,31,32], here we present the variability of the ellipsometric angle Δ as a function of temperature for one selected wavelength. In this case, λ = 900 nm, due to the low dispersion of the measured points. Ellipsometric temperature scans were obtained based on about 200 measurements, performed one after the other, every 10 s, with a successive increase in the temperature value.

The temperature scans for studied samples are presented in Figure 5a–j. In order to facilitate the comparison of the results obtained with both methods, the temperature scans were shown in pairs for VTSE and DSC, and successively for layers with different mixture compositions.

As can be seen, most thermal transitions can be detected by both methods, and the corresponding characteristic temperatures are consistent with each other. Therefore, when identifying the nature of a specific thermal transition, we relied mainly on the DSC results. With regard to PDPP4T in Figure 5a,b, we can detect four characteristic temperatures from the slope change Δ(T): approximately 32, 69, 129 (glass transition—T_g1_–T_g3_) and 247 °C (cold crystallization temperature—T_cc_). In turn, the thermal changes visible on the DSC charts correspond to temperatures of about 132 (T_g_), 307 (melting point—T_m1_) and 330 °C (T_m2_) (for the scratched film). In contrast, the measurement on the original powder material shows thermal changes at about 69 (Tg), 330 °C (Tm) for the heating cycle and a very pronounced cold crystallization peak at about 290 °C in the cooling scan. It should be noted that more thermal transitions can be observed in the films than in the powder material. This is because the internal structure of the solvent treated material and the spin coating process leads to a better order than that of the powdered material. The crystallization in PDPP4T film occurs around 247 °C. In Figure 5c,d, for PDBPyBT, four characteristic temperatures are seen at 25, 57, 162 (T_g1_–T_g3_) and 268 °C (T_cc_) in the Δ(T) plot. The corresponding temperatures in the DSC plot are shown at around 32, 162 (T_g1_–T_g2_) and 257 °C (T_cc_) in the heating scan for the scratched film, and at around 162 (T_g_) and 261 °C (T_cc_) in the heating scan taken for the powdered material. The results taken on the 50:50% blend are shown in Figure 5e,f. Three characteristic temperatures detected in the ellipsometric temperature scan are located at around 77, 125 (T_g1_–T_g2_) and 244 °C (T_cc_); the corresponding temperatures in DSC curves are around 61, 148 (T_g1_–T_g2_) and 250 °C (T_cc_). The characteristic temperature visible at 322 °C corresponds to the melting point. In turn, the thermal transition temperatures for the PDPP4T (75%):PDBPyBT (25%) blend are shown in Figure 5g,h. The first three temperatures (36, 74 and 110 °C) correspond to glass transitions in the layer. The thermal transition at 198 °C is interpreted as the cold crystallization, whereas, the temperature of 323 °C is the melting point of the film. Finally, the thermal transition temperatures for the PDPP4T (25%):PDBPyBT (75%) blend film are shown in Figure 5i,j. Two glass transitions occur at around 80 and 145 °C, and a clear crystallization peak is also visible at 270 °C in the DSC cooling scan.

All identified characteristic temperatures were plotted on a phase diagram which shows the temperature values of individual thermal transitions for the mixtures depending on the percentage of PDBPyBT content. The phase diagram thus obtained is shown in Figure 6. All added melting points were derived from the DSC plots. There are two melting points for the pure PDPP4T film. In the layers of pure polymer (one-component), as well as in the layers of their blends, the presence of several glass transition temperatures is visible. This observation suggests that we are dealing with single-phase samples and that these polymers mix well with each other in a wide range of compositions. On the other hand, the presence of several glass transitions in the examined layers is probably caused by the collective movements of various segments of the polymer chain; for example, aromatic rings in the backbone or π conjugated segments, as well as the aliphatic side chains of the test compounds interacting differently.

### 3.4. AFM Results

The surface morphology of PDPP4T and PDBPyBT is presented in Figure 7a–d. The morphologies of blends are shown in Figure 7e–j. The surface of PDPP4T seems to be more amorphous than the surface of PDBPyBT, with single conical crystallites of 0.3–0.6 μm diameter and 30–125 nm height. The surface of PDBPyBT is more ordered. The crystalline domains are visible in the 1 × 1 μm sale. In addition, the single conical crystallites are present on the sample surface, which their diameter within 0.6–0.8 μm and their height around 53–166 nm (10 × 10 μm scale). The appearance of the surface indicates the liquid crystalline nature of the tested material. The remaining topographies show that the surface is more or less ordered, depending on the PDBPyBT content. The surface morphology of PDPP4T (50%):PDBPyBT (50%) seems to be most regular, with single conical crystallites (diameter of around 0.25 μm, height of around 20 nm). The surface of the PDPP4T (75%):PDBPyBT (25%) blend is more similar to the surface morphology of the PDPP4T sample. This is caused by the higher content of this material. The diameter of single crystals on the surface is within 0.25–1.25 μm and their height is in the range of 12–45 nm. The surface of PDPP4T (25%):PDBPyBT (75%) is more ordered, also with single conical crystals. Their diameters are in the range of 0.3–0.6 μm and their height is in range of 8–20 nm. The most regular crystalline domains are visible on the surface of the PDPP4T (50%):PDBPyBT (50%) blend film. This fact indicates a more anisotropic structure of the tested films. This is consistent with the microscopic image disclosed from the XRD results, which showed that the polymer chains in the tested layers are positioned parallel to the substrate surface in edge-on orientation [56].

The root mean square roughness—*Rq* was used to characterize the surface all of the tested samples. It is defined as [61]:(4)$$Rq=\sqrt{\frac{1}{m}\sum_{i=1}^{m}{\left(Z_{i}-\bar{Z}\right)}^{2}},\;$$
where *m* is the number of sampled points, *Z_i_* is the height of each point and $\bar{Z}$ is the mean height value [61]. *Rq* was determined for three surface sizes: 1 × 1, 2 × 2 and 10 × 10 μm, and is shown in Figure 8.

As can be seen in Figure 8, the *Rq* value for AFM images at the 10 × 10 scale is lower for the polymer blend film. However, in the case of pure polymers, *Rq* is clearly higher. This indicates that the surface of the mixture layers was flatter.

## 4. Conclusions

We presented research on the influence of the composition of PDPP4T:PDBPyBT blends on their physical properties, focusing on thermal transitions. The conducted research included XRD diffractometry, spectroscopic ellipsometry, variable-temperature ellipsometry and atomic force microscopy. Using VTSE and DSC techniques, we observed numerous thermal changes in the PDPP4T, PDBPyBT layers and their mixtures, and we determined their characteristic temperatures. Our approach allowed us to construct a detailed phase diagram of the PDPP4T:PDBPyBT films depending on the PDBPyBT content. The samples tested as layers were found to have more thermal transitions than the original powder material. The values of glass transition and crystallization temperatures are the highest at 50% concentration of PDPP4T and PDBPyBT, and for a higher content of PDBPyBT in the mixtures. The results of thermal analysis and X-ray diffraction indicate that the polymers PDPP4T and PDBPyBT mix well and form single-phase layers. Moreover, atomic force microscopy showed the presence of conical crystals on the polymer surface, and the RMS roughness is the lowest for a 50% mixture of both components. On the other hand, the optical results showed that the value of the optical energy gap of PDPP4T compared to the PDPP4T:PDBPyBT blends did not change significantly at about 1.4 eV. Finally, it should be emphasized that the obtained layers are characterized by the edge-on arrangement of the polymer chains, which is advantageous for OFET devices.

## Figures and Tables

**Figure 1 materials-15-08392-f001:**
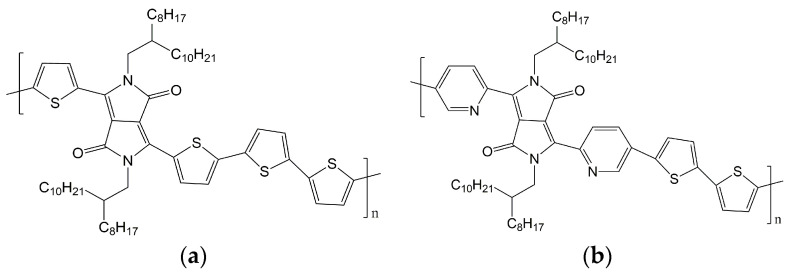
Chemical structures of PDPP4T (**a**) and DPPDPyBT (**b**).

**Figure 2 materials-15-08392-f002:**
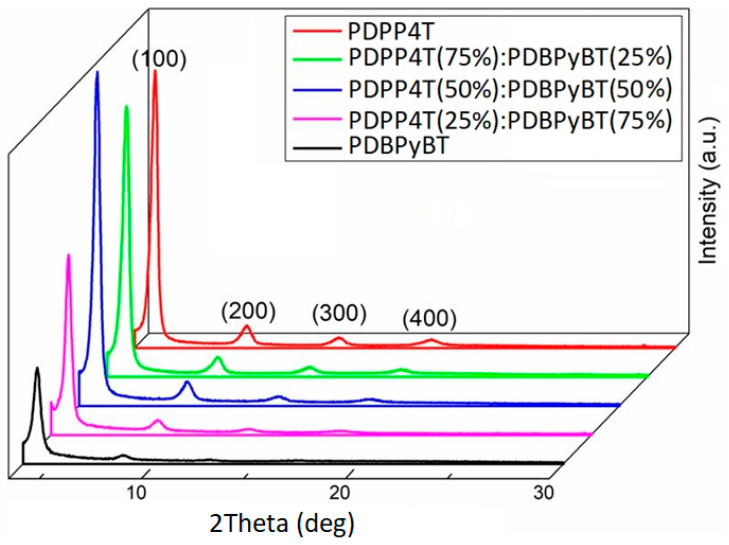
XRD scans, in coupled two-theta/theta mode, of PDPP4T, PDBPyBT and their blends.

**Figure 3 materials-15-08392-f003:**
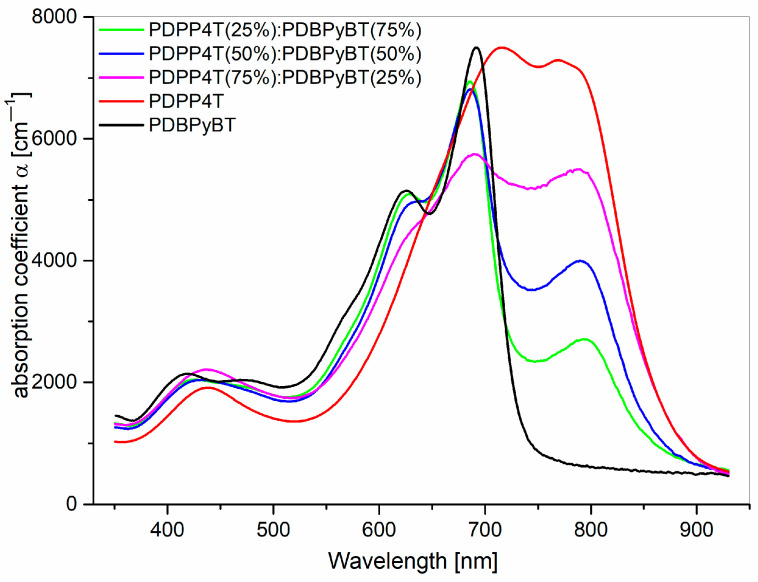
Absorption coefficient of PDPP4T, PDBPyBT films and their blends in the spectral range of visible light.

**Figure 4 materials-15-08392-f004:**
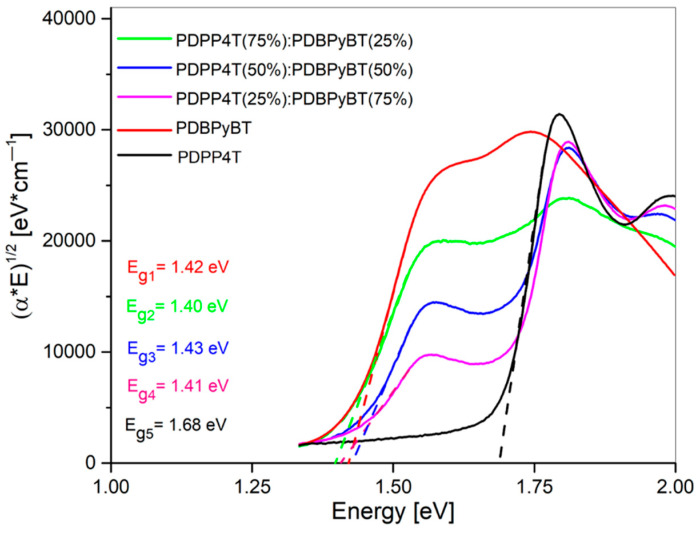
Bandgap Eg determination for PDPP4T, PDBPyBT and its mixtures.

**Figure 5 materials-15-08392-f005:**
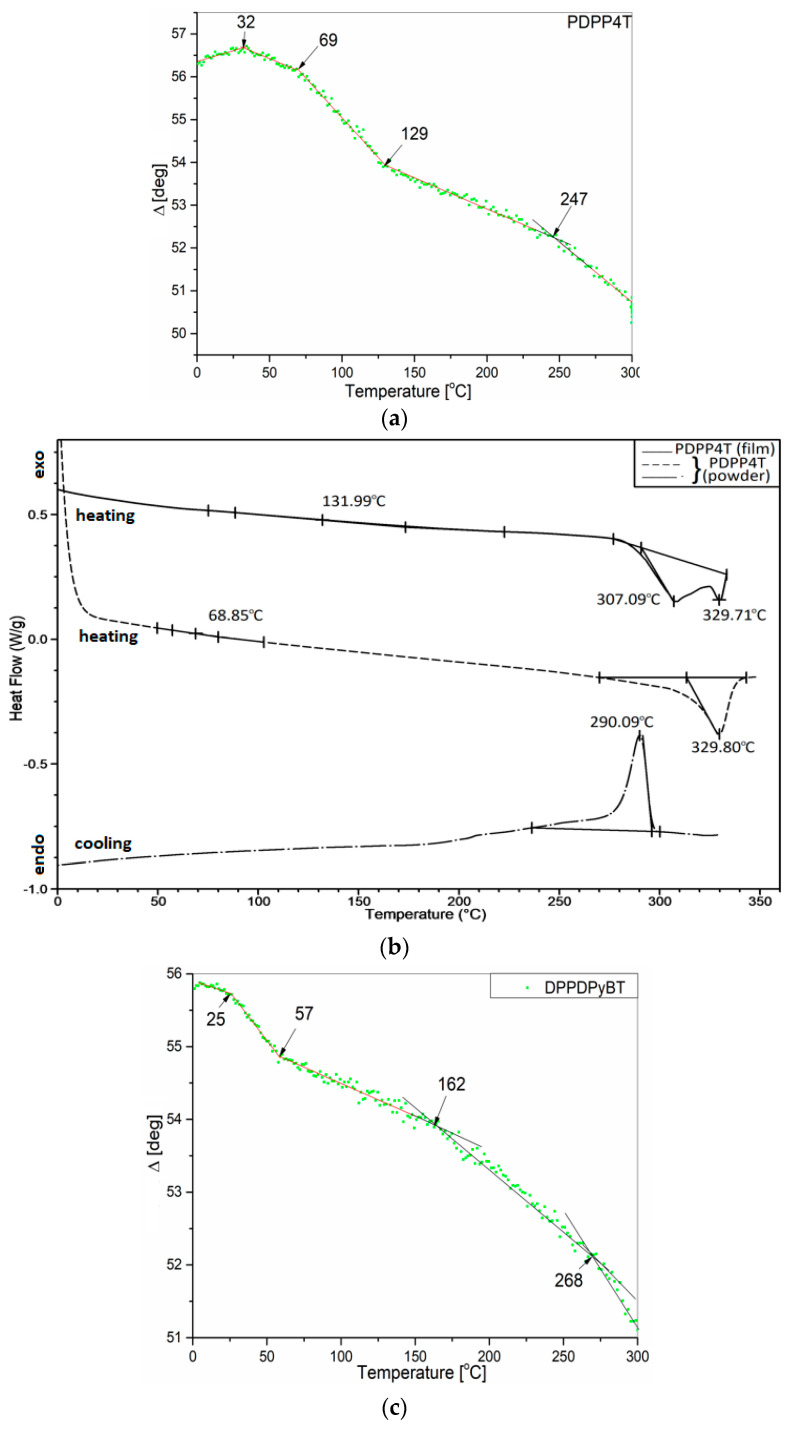
Ellipsometric angle Δ at 900 nm as a function of temperature and DSC plots for PDPP4T (**a**,**b**), PDBPyBT (**c**,**d**) and their blends—PDPP4T (50%):PDBPyBT (50%) (**e**,**f**), PDPP4T (75%):PDBPyBT (25%) (**g**,**h**), PDPP4T (25%):PDBPyBT (75%) (**i**,**j**).

**Figure 6 materials-15-08392-f006:**
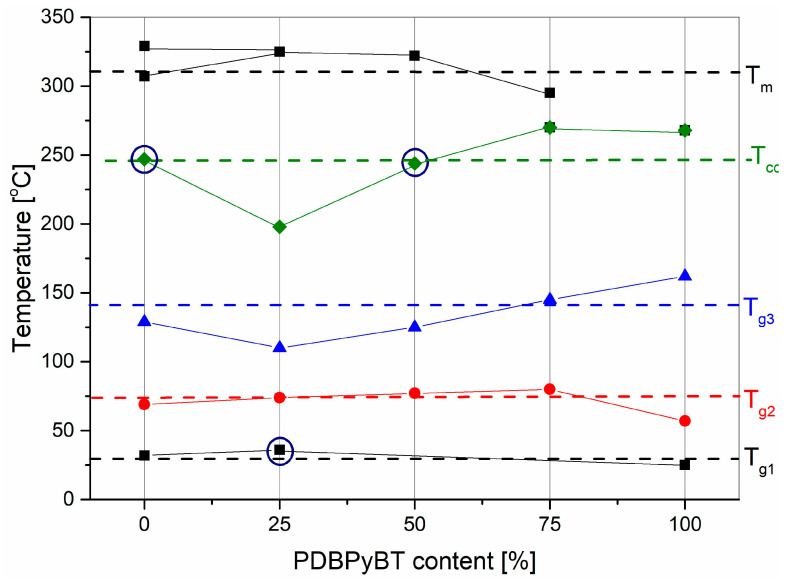
Phase diagram of PDPP4T:PDBPyBT blend films, prepared based on DSC and temperature ellipsometry. Thermal transitions marked with a dark blue circle were detected only in VTSE measurements. T_g_ is the glass transition, T_cc_ is the cold crystallization temperature and T_m_ is the melting point.

**Figure 7 materials-15-08392-f007:**
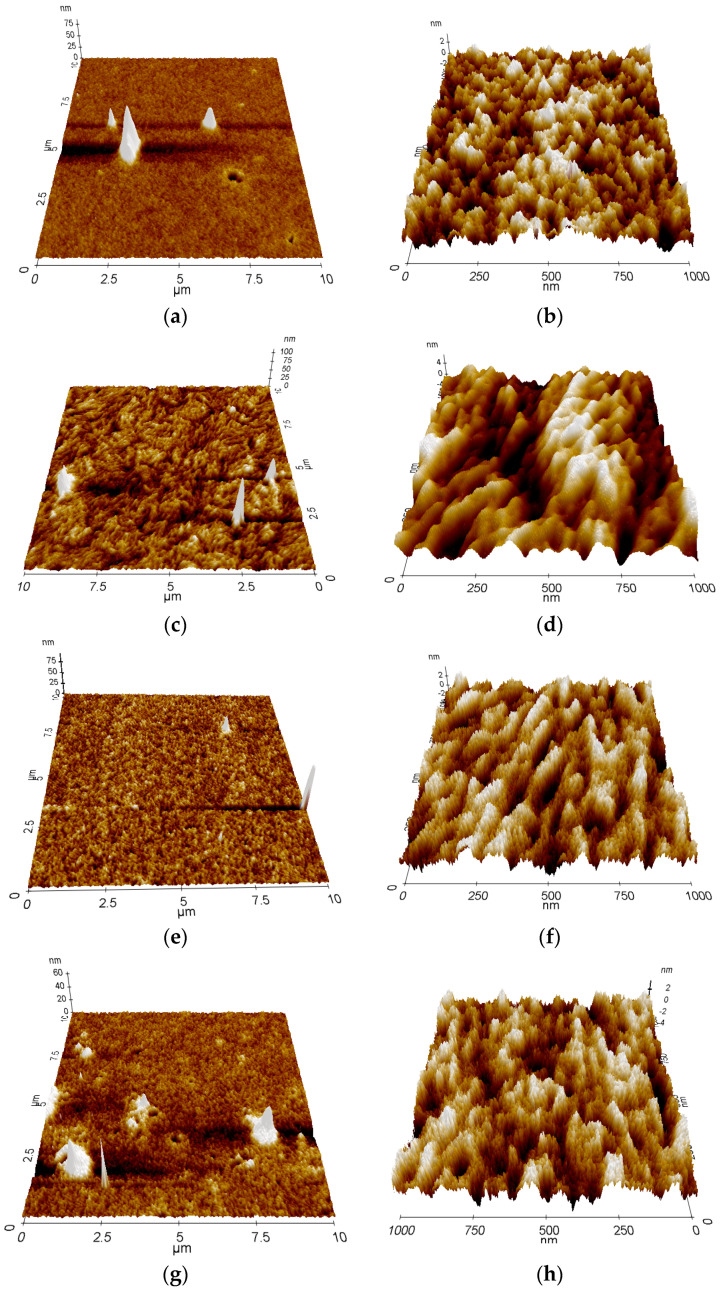
AFM 10 × 10 μm and 1 × 1 μm 3D topographic images of surface of PDPP4T (**a**,**b**), PDBPyBT (**c**,**d**) and their blends—PDPP4T (50%):PDBPyBT (50%) (**e**,**f**), PDPP4T (75%):PDBPyBT (25%) (**g**,**h**), PDPP4T (25%):PDBPyBT (75%) (**i**,**j**).

**Figure 8 materials-15-08392-f008:**
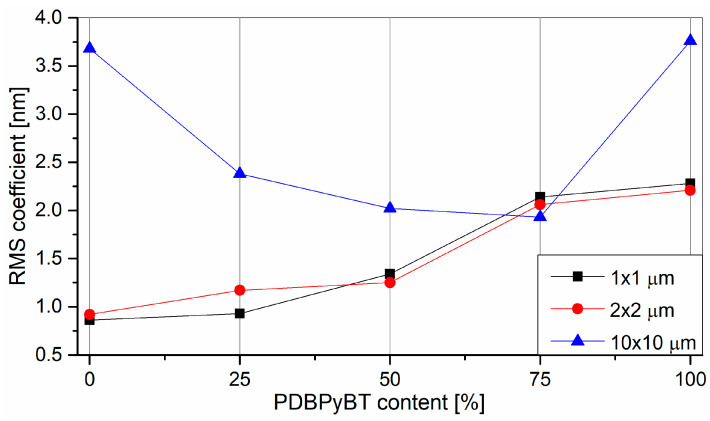
The mean square root of the roughness of the tested layers.

**Table 1 materials-15-08392-t001:** Summary of the spin coating parameters and composition of polymer solutions.

Material Content [%]	Spinning Rate [rpm]	Spinning Time [s]	Atmosphere
PDPP4T (100)	3000	60	air
PDPP4T (25):PDBPyBT (75)			
PDPP4T (50):PDBPyBT (50)			
PDPP4T (75):PDBPyBT (25)			
PDBPyBT (100)			

**Table 2 materials-15-08392-t002:** Thickness of layers deposited on silicone and glass substrates.

Material Content [%]	Thickness [nm] of Layer Deposited on:	
	Si/SiOx Substrates	Microscopic Glass
PDPP4T (100)	717	365
PDPP4T (25):PDBPyBT (75)	758	200
PDPP4T (50):PDBPyBT (50)	914	202
PDPP4T (75):PDBPyBT (25)	599	197
PDBPyBT (100)	676	155

## Data Availability

Pre-print: http://doi.org/10.2139/ssrn.4209016.

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
