# Peer review of "An Investigation of the Thermal Transitions and Physical Properties of Semiconducting PDPP4T:PDBPyBT Blend Films"

_materials, 2022, doi:10.3390/ma15238392_

Round 1

Reviewer 1 Report

See the attachment

Author Response

We sincerely thank all the reviewers for their accurate and valuable comments. We tried to take all of them into account and answer all of them.

Reviewer 1

The work is of interest in the theoretical and applied field of organic semiconductors. I

recommend the publication of this work after a considerable improvement of it.

- I recommend improving the writing:

in pag 2. 6.3 cm2*V-1*s-1 50 and 2.78 cm2*V-1*s-1 respectively) [14-15]. (removing

the * symbol or replacing it with a dot.);

-fixed

- In Chapter 1, the full names of the abbreviations must be given („using VTSE and DSC

measure-68 ment techniques.”) - these are given in the following chapters.

-fixed

- The preparation method is mentioned in the text (homogenizer at 16 kJ for 10 minutes.)

and in Table 1, 60 s is specified; missing information about: treatment atmosphere and

heating and cooling speed?

-fixed

- It is recommended to complete the table with all the prepared samples (nr. crt., noting

the  samples  that  reflect  the  concentrations,  the  substrate  used,  etc.);  In  Table  1  the

marking  of  samples  is  for  example:  PDPP4T:PDBPyBT  25%/75%,  In  Table  2  is

PDPP4T:PDBPyBT 25/75% and in Figure 2 it is PDPP4T(75%):PDBPyBT (25%) and at

last „PDPP4TxPDBPyBT(1-x) films for five composition x=1.0, 0.75,0.5, 0.25 and 0.0.”

-fixed

- It is recommended to give more details about the substrates of the samples;

-fixed

- For the crystallographic planes - in the figure there are round brackets (hkl) and in the

text  there  are  square  brackets  [  ];  the  units  of  measure  in  the  graphs,  either  in  round

brackets or in square brackets. In Figure 2 "2 Theta" is written and in some parts of the

text "2Θ  " is  written; 

„Figure 7. AFM  10x10 μm  topographic images of surface” 2D..

3D..?

-fixed

- The n or p semiconductor types are discussed in abstract - more details should be given

in the text of the paper.

-fixed

- Subchapter 3.3. must be detailed. What is followed, what measurements are made and

what do these measurements represent?

-Fixed, several sentences with description has been added into the text.

-In Subchapter 3.5, the table is not noted and is

not clearly represented.

-fixed

- The conclusions must more clearly reflect the content of the work.

-fixed

Reviewer 2 Report

In this manuscript, the phase thermal transitions and physical properties of PDPP4T:PDBPyBT blend have been investigated with the help of VTSE, DSC and AFM. The topic itself is important, especailly in the field of semiconducting devices. The manuscript, however, has not been organized well.

1. The authors should check the whole manuscript carefully. Some expression can be corrected. For instance, in Line 25, "The AFM microscopy showed ****" AFM means atomic force microscopy, what is the meaning of AFM microscopy?

2. Several substrates have been employed in this work. What is the difference between them? How about the influcence of substrate on DSC, AFM results?

3. It is necessary to give more details in experimental section, e.g., heating/cooling rate during DSC measurement.

4. How about the interaction between PDPP4T and PDBPyBT?  Is there phase separation between them? How about the influence of crystallziation on phase separation?

5. In DSC results (Fig. 5), how to define the characteristic temperatures? e.g., 32, 69, 129, 247 in Fig. 5a? How to get the green curve in Fig. 5?

6. There are solid curve and dash curve in Fig. 5. The former represents the results from film while the latter corresponds to powder. How to perform DSC meansurement on film? More details and description are necessary.

7. In AFM results, how to distinguish conical crystallites from impurity?

8. In AFM results of neat PDPP4T and neat PDBPyBT, there are conical crystallites. In Fig. 7c and 7e, however, the surface becomes flat. What is the reason?

9. It would be better to show AFM 3D images in the same height scale.

10. How to compare DSC with AFM results? The former and the latter focus on bulk and film surface respectively?

Author Response

We sincerely thank all the reviewers for their accurate and valuable comments. We tried to take all of them into account and answer all of them.

Reviewer 2

In this manuscript, the phase thermal transitions and physical properties of PDPP4T:PDBPyBT blend have been investigated with the help of VTSE, DSC and AFM. The topic itself is important, especailly in the field of semiconducting devices. The manuscript, however, has not been organized well.

-manuscript has been carefully checked and text has been corrected

1. The authors should check the whole manuscript carefully. Some expression can be corrected. For instance, in Line 25, "The AFM microscopy showed ****" AFM means atomic force microscopy, what is the meaning of AFM microscopy?

-fixed

2. Several substrates have been employed in this work. What is the difference between them? How about the influcence of substrate on DSC, AFM results?

-We have used several kinds of substrates, however in the case of DSC, materials were scratched out of substrates and measured in powder form

3. It is necessary to give more details in experimental section, e.g., heating/cooling rate during DSC measurement.

-fixed

4. How about the interaction between PDPP4T and PDBPyBT?  Is there phase separation between them? How about the influence of crystallziation on phase separation?

- In the revised manuscript, we clearly explained that PDPP4T and PDBPyBT blend well within the wide composition range studied. In fact, we do not see phase separation in the layers of the blend. As for the interactions between these polymers, their presence is reflected in changes in the value of the glass transition temperature with the change in the composition of the blend.

  1. In DSC results (Fig. 5), how to define the characteristic temperatures? e.g., 32, 69, 129, 247 in Fig. 5a? How to get the green curve in Fig. 5?

-The ellipsometric temperature curves are very good and easy way to find values of thermal transitions. The DSC method was used to identify the character of detected thermal transitions. The method of getting the green curve has been described in text.

6. There are solid curve and dash curve in Fig. 5. The former represents the results from film while the latter corresponds to powder. How to perform DSC meansurement on film? More details and description are necessary.
-Fixed. More details has been added in text.

  1. In AFM results, how to distinguish conical crystallites from impurity?
  2. In AFM results of neat PDPP4T and neat PDBPyBT, there are conical crystallites. In Fig. 7c and 7e, however, the surface becomes flat. What is the reason?
  3. It would be better to show AFM 3D images in the same height scale.
    10. How to compare DSC with AFM results? The former and the latter focus on bulk and film surface respectively?

- Thermal analysis shows very clearly that in the prepared layers the phenomenon of cold crystallization takes place, while the results of X-ray diffraction indicate the clearly crystalline nature of the tested materials. Based on this attitude, we believe that the AFM images show crystallites, not impurities.

Reviewer 3 Report

Authors present a study on thermal transitions and physical properties of semiconducting PDPP4T:PDBPyBT blend films.

The manuscript lacks of several work and discussion, and thus, in my opinion, it can not be considered for publication in the present form.

Here is a list of queries for authors to consider for future submissions:

Q1) The DSC methodology section needs more details such as cooling and heating rates.

Q2) The discussion of XRD results is quite poor and needs to be referenced with other similar works in the literature. 

Q3) DSC data displayed in Figure 5 must be merged in a single figure, the heatflow (exo and endo) must be included in the Y-axis and the cooling/heating curves must be labeled as well.

Q4) Some phase transitions, particularly glass transitions and melting points are not clearly identified and additional thermogravimetric analyses are needed to discard other endothermic processes related to partial decompositions.

Q5) The crystallite sizes obtained are quite large and it is not clear how authors have estimated them. Here, a comparisson with those estimated using the Scherrer equation from XRD analyses is needed.

Author Response

We sincerely thank all the reviewers for their accurate and valuable comments. We tried to take all of them into account and answer all of them.

Reviewer 3

Authors present a study on thermal transitions and physical properties of semiconducting PDPP4T:PDBPyBT blend films.

The manuscript lacks of several work and discussion, and thus, in my opinion, it can not be considered for publication in the present form.

Here is a list of queries for authors to consider for future submissions:

Q1) The DSC methodology section needs more details such as cooling and heating rates.

-fixed

Q2) The discussion of XRD results is quite poor and needs to be referenced with other similar works in the literature.

-fixed

Q3) DSC data displayed in Figure 5 must be merged in a single figure, the heatflow (exo and endo) must be included in the Y-axis and the cooling/heating curves must be labeled as well.

-fixed

Q4) Some phase transitions, particularly glass transitions and melting points are not clearly identified and additional thermogravimetric analyses are needed to discard other endothermic processes related to partial decompositions.

- Within the time frame we got from the editor, such a measurement is technically impossible for us.

Q5) The crystallite sizes obtained are quite large and it is not clear how authors have estimated them. Here, a comparisson with those estimated using the Scherrer equation from XRD analyses is needed.

-The crystallite size using Scherrer equation from XRD has been estimated in the revised manuscript.

Reviewer 4 Report

Recommendation

Major Revision

Comments:

This manuscript describes the thermal and physical properties of polymeric thin films based on semiconducting PDPP4T/ PDBPyBT polymer blends. The physical properties and thermal transitions of PDPP4T:PDBPyBT blend films have been characterized by XRD diffractometry, spectroscopic ellipsometry, variable temperature ellipsometry, atomic force microscopy and electrical measurements. A phase diagram for PDPP4T: PDBPyBT blends have been constructed on the base of variable temperature spectroscopic ellipsometry and differential scanning calorimetry. The AFM microscopy showed coefficient value of RMS roughness coefficient of blends is lower than that for pure materials. The electrical measurements showed that open circuit voltage (Voc) value of blend is higher than for pure materials. It is a systematic study of DPP-based donor/acceptor blend films and provides a perspective solution in constructing all-polymer solar cells. However, the analysis of the data in this manuscript is insufficient to draw a conclusion on how the ratio of components influence the physical and thermal properties of D/A blends. The authors just extracted the parameters from the spectra and listed in the context without much comparison and discussion. Moreover, some very important features of D/A polymers such as electrochemical properties, as well as the parameters for OPVs such as Jsc, FF and PCE in addition to Voc, need to be provided. The following questions need to be tackled:

1. Please provide the parameters of Joc, FF, PCE and EQE in addition to Voc for the OPV devices with varied component ratios.

2. Please provide the redox potentials and energy levels of the donor and acceptor polymers estimated by cyclic voltammetry (CV) or Differential pulse voltammetry (DPV).

3. On page 4, the authors wrote: “These values, according to Bragg's 144 law for d(100), correspond to the following lengths of the parameter a: respectively 19.6, 145 19.8, 20.0, 20.2 and 21.0 Å for the film with x = 0.0, 0, 25, 0.5, 0.75 and 1.0. Summarizing, 146 these results clearly show that the examined layers contain a structural arrangement of 147 polymer chains, the so-called stacking.” Please clarify what did these x-ray data indicate about the stacking modes of polymer films. What does the term “lengths” refer to here, such as coherence length or π-π stacking distance? What is the specific structural arrangement, such as face-to-face, edge-to-face?

4. Are the films used for the absorption spectra in Figure 3&4 and the thickness test in Table 2 the same ones? If so, then the results for the relative absorption values and the thickness data are not consistent? For instance, the absorption of PDPP4T:PDBPyBT 50/50% film in the range of 750-900 nm should be ca. 60% of that compared to the absorption of pure PDBPyBT film. If not, please just normalized the data for clarity. Please check the data.

5. Please make comparisons between the thermal properties of films with different D/A ratios to clarify how the components of blend films influence their thermal properties.

6. Lines 81&84 on page 2: the abbreviation names of the acceptor polymer should be unified to be either PDBPyBT or DPPDPyBT.

7. In Figure 1, the bond widths and the fonts in the structural drawings should be unified.

8. Line 130 on page 3: “Fig. 1” should be changed to “Figure 2”.

9. What does Voc refer to when talking about the devices with neat film of either donor or acceptor components? Where is the Voc originated for neat films? Please explain.

Author Response

We sincerely thank all the reviewers for their accurate and valuable comments. We tried to take all of them into account and answer all of them.

Recommendation

Major Revision

Comments:

Manuscript ID: materials-1997967

Title:  The  investigation  on  thermal  transitions  and  physical  properties  of

semiconducting PDPP4T:PDBPyBT blend films

Barbara  Hajduk  *  ,  PaweÅ‚  Jarka  *  ,  Tomasz  TaÅ„ski  ,  Henryk  Bednarski  ,  Henryk

Janeczek , Paweł Gnida , Mateusz Fijałkowski

This manuscript describes the thermal and physical properties of polymeric thin films

based on semiconducting PDPP4T/ PDBPyBT polymer blends. The physical properties

and thermal transitions of PDPP4T:PDBPyBT blend films have been characterized by

XRD  diffractometry,  spectroscopic  ellipsometry,  variable  temperature  ellipsometry,

atomic force microscopy and electrical measurements. A phase diagram for PDPP4T:

PDBPyBT  blends  have  been  constructed  on  the  base  of  variable  temperature

spectroscopic ellipsometry and differential scanning calorimetry. The AFM microscopy

showed coefficient value of RMS roughness coefficient of blends is lower than that for

pure materials. The electrical measurements showed that open circuit voltage (Voc)

value of blend is higher than for pure materials. It is a systematic study of DPP-based

donor/acceptor  blend  films  an  provides  a  perspective  solution  in  constructing  all-

polymer solar cells. However, the analysis of the data in this manuscript is insufficient

to draw a conclusion on how the ratio of components influence the physical and thermal

properties of D/A blends. The authors just extracted the parameters from the spectra

and listed in the context without much comparison and discussion. Moreover, some

very important features of D/A polymers such as electrochemical properties, as well as

the  parameters  for  OPVs  such  as  Jsc,  FF  and  PCE  in  addition  to  Voc,  need  to  be

provided. The following questions need to be tackled: 1. Please provide the parameters of Joc, FF, PCE and EQE in addition to Voc for the

OPV devices with varied component ratios.

  1. Please provide  the  redox  potentials  and  energy  levels  of  the  donor  and  acceptor

polymers  estimated  by  cyclic  voltammetry  (CV)  or  Differential  pulse  voltammetry

(DPV).

Within the time frame we have received from the editorial office, such a measurement is technically impossible for us. For this reason, we have provided only references to the literature [60 - 61].

  1. On page 4, the authors wrote: “These values, according to Bragg's 144 law for d(100),

correspond to the following lengths of the parameter a: respectively 19.6, 145 19.8,

20.0, 20.2 and 21.0 Å for the film with x = 0.0, 0, 25, 0.5, 0.75 and 1.0. Summarizing,

146  these  results  clearly  show  that  the  examined  layers  contain  a  structural

arrangement of 147 polymer chains, the so-called stacking.” Please clarify what did

these x-ray data indicate about the stacking modes of polymer films. What does the

term “lengths” refer to here, such as coherence length or π-π stacking distance? What

is the specific structural arrangement, such as face-to-face, edge-to-face?

-Whole paragraph has been improved taking into account raised issues. It is also clarified that structural arrangement of polymer chains is edge-on.

  1. Are the films used for the absorption spectra in Figure 3&4 and the thickness test in

Table 2 the same ones? If so, then the results for the relative absorption values and the

thickness data are not consistent? For instance, the absorption of PDPP4T:PDBPyBT

50/50% film in the range of 750-900 nm should be ca. 60% of that compared to the

absorption of pure PDBPyBT film. If not, please just normalized the data for clarity.

Please check the data.

-Thank you for this attention. At this point there was a mistake and the thicknesses of the layers on the silicon were used instead of the thicknesses of the layers deposited on the glass for which the transmission was measured. The thicknesses of these layers were added to the table and used for the calculation of the optical absorption coefficient and determination of individual energy gaps.

  1. Please make comparisons between the thermal properties of films with different D/A

ratios to clarify how the components of blend films influence their thermal properties.

-The summary of these thermal properties is presented in phase diagram. The thermal transitions were marked using spectroscopic ellipsometry and defined using differential scanning calorimetry.

  1. Lines 81&84 on page 2: the abbreviation names of the acceptor polymer should be

unified to be either PDBPyBT or DPPDPyBT.

-Fixed

  1. In Figure 1, the bond widths and the fonts in the structural drawings should be unified.

-Fixed

  1. Line 130 on page 3: “Fig. 1” should be changed to “Figure 2”.

-Fixed

  1. What does Voc refer to when talking about the devices with neat film of either donor

or acceptor components? Where is the Voc originated for neat films? Please explain.

-As it turned out that the microstructure of these layers is not suitable for OPV, we removed the paragraph on electrical measurements.

Reviewer 5 Report

B. Hajduk et al. have investigated the thermal transitions and physical properties of donor-acceptor semiconducting-based PDPP4T:PDBPyBT blend films. The blend materials' structure, morphology, and thermal properties were characterized and the manuscript is quite interesting. The authors need to revise this manuscript satisfactorily according to the following comments before it accepts for publication in Materials.

1. Why did the authors chosen these PDPP4T and DPPDPyBT donor-acceptor polymers for this study? The authors need to discuss the material background in the introduction part.

2. The resolution of all figures provided in the manuscript is low. All the figures need to be increased the resolution, especially Fig 5.

3. The multiple times repeated DSC caption should be removed from Figure 6. Better to mention the legend one time with a symbol denoting curves.

4. The authors have discussed the optical bandgap from the absorption spectra of blends and it's not sufficient; the discussion of electronic transition should be included.

5. Did the author see any interaction like inter/intramolecular charge transfer of blends? This needs to be addressed by using supporting FTIT/absorption spectra.

Author Response

We sincerely thank all the reviewers for their accurate and valuable comments. We tried to take all of them into account and answer all of them.

Reviewer 5

  1. Why did the authors chosen these PDPP4T and DPPDPyBT donor-acceptor polymers for this study? The authors need to discuss the material background in the introduction part.

-fixed

  1. The resolution of all figures provided in the manuscript is low. All the figures need to be increased the resolution, especially Fig 5.

-fixed

  1. The multiple times repeated DSC caption should be removed from Figure 6. Better to mention the legend one time with a symbol denoting curves.

-fixed

  1. The authors have discussed the optical bandgap from the absorption spectra of blends and it's not sufficient; the discussion of electronic transition should be included.

-fixed

  1. Did the author see any interaction like inter/intramolecular charge transfer of blends? This needs to be addressed by using supporting FTIT/absorption spectra.

-Within the time frame we have received from the editorial office, it is technically impossible for us to perform such additional research.

Round 2

Reviewer 1 Report

See attachment

Author Response

We sincerely thank again all the reviewers, for their accurate and valuable comments. We tried to take all of them into account and answer all of them.

Reviewer 1

Reviewer 1

11.10.2022

Materials

Manuscris; 1997967

Title: The investigation on thermal transitions and physical proper-ties of semiconducting

PDPP4T:PDBPyBT blend films

Dear Editor

The authors improved the paper but not enough:

- All references are to Chapter 1, the other chapters have no new references.

- Ref. 51-59 missing in the text ??

-fixed

- for formula (1), citation is recommended - as well as for formula (2)

-fixed

- „(lamellar spacing around 21 Å) [52-53].” but it is not 51 nor 52-59

-fixed

- „….using Cauchy model [29-30].” to check

 -fixed

Best regards,

Reviewer 2 Report

According to the comments, the authors made improvement. Now, it can be published.

Author Response

We sincerely thank again all the reviewers, for their accurate and valuable comments. We tried to take all of them into account and answer all of them.

Reviewer 3 Report

Although not all queries were completely solved, I think the manuscript has drastically improved compared to the original version to be worth for publication.

Author Response

(The authors gave the same response as above.)

Reviewer 4 Report

The manuscript has been sufficiently improved to warrant publication in Materials. 

Author Response

(The authors gave the same response as above.)

Reviewer 5 Report

The authors have addressed all the reviewers comments satisfactorily, and hence I would recommend this paper for publication in its current form.

Author Response

(The authors gave the same response as above.)
